# Anodal-TDCS over Left-DLPFC Modulates Motor Cortex Excitability in Chronic Lower Back Pain

**DOI:** 10.3390/brainsci12121654

**Published:** 2022-12-02

**Authors:** Emily J. Corti, An T. Nguyen, Welber Marinovic, Natalie Gasson, Andrea M. Loftus

**Affiliations:** 1School of Population Health, Curtin University, GPO Box U1987, Bentley, WA 6102, Australia; 2Curtin Neuroscience Research Laboratory, Curtin University, GPO Box U1987, Bentley, WA 6102, Australia

**Keywords:** motor cortex, excitability, transcranial magnetic stimulation, transcranial direct current stimulation, chronic lower back pain

## Abstract

Chronic pain is associated with abnormal cortical excitability and increased pain intensity. Research investigating the potential for transcranial direct current stimulation (tDCS) to modulate motor cortex excitability and reduce pain in individuals with chronic lower back pain (CLBP) yield mixed results. The present randomised, placebo-controlled study examined the impact of anodal-tDCS over left-dorsolateral prefrontal cortex (left-DLPFC) on motor cortex excitability and pain in those with CLBP. Nineteen participants with CLBP (Mage = 53.16 years, SDage = 14.80 years) received 20-min of sham or anodal tDCS, twice weekly, for 4 weeks. Short interval intracortical inhibition (SICI) and intracortical facilitation (ICF) were assessed using paired-pulse Transcranial Magnetic Stimulation prior to and immediately following the tDCS intervention. Linear Mixed Models revealed no significant effect of tDCS group or time, on SICI or ICF. The interactions between tDCS group and time on SICI and ICF only approached significance. Bayesian analyses revealed the anodal-tDCS group demonstrated higher ICF and SICI following the intervention compared to the sham-tDCS group. The anodal-tDCS group also demonstrated a reduction in pain intensity and self-reported disability compared to the sham-tDCS group. These findings provide preliminary support for anodal-tDCS over left-DLPFC to modulate cortical excitability and reduce pain in CLBP.

## 1. Introduction

Chronic Lower Back Pain (CLBP) is a leading cause of disability worldwide and is the most common job-related disability and cause of work absence [1]. Despite affecting half a billion people world-wide, little is understood about the cause of CLBP, and commonly prescribed treatments (including opioids) are often ineffective for the long-term management of CLBP [2,3]. Research has recently begun to consider the potential for nonpharmacological management of CLBP.

Research suggests that cortical adaptations to pain may contribute to its persistence [4,5,6]. A number of studies indicate decreased activation in areas of the pain ‘Neuromatrix’, which includes the motor cortex and prefrontal cortex [7,8,9,10]. Chronic pain, including CLBP, is associated with altered excitability in the motor cortex [5,6,11]. Research in experimentally induced and chronic pain have suggested that short interval intracortical inhibition (SICI) and intracortical facilitation (ICF) may be the key mechanisms associated with the maintenance of pain [12], whereby an imbalance between inhibition (GABAA; indicated by SICI) and excitation (glutamate indicated by ICF) may be associated with increased pain intensity [5,11,13]. CLBP imaging studies also report reduced cortical grey matter density in the dorsolateral prefrontal cortex (DLPFC) [14,15]. In light of these findings, emerging research has sort to establish whether chronic pain can be managed using non-invasive brain stimulation to modulate activity in the brain areas involved in pain processing [16,17].

Transcranial Direct Current Stimulation (tDCS) is a non-invasive brain stimulation technique that delivers low intensity electrical currents to modulate neural activity [18]. tDCS has the potential to change chronic pain by acting upon the endogenous opioid system, changing the emotional appraisal of pain, and altering the pain signal via descending pathways [19,20]. Most studies examining tDCS in chronic pain have focused on stimulation of the motor cortex. In therapy-resistant chronic pain syndromes, such as post-stroke pain, back pain, and fibromyalgia, anodal-tDCS (a-tDCS) over the motor cortex increased intracortical excitability and decreased pain ratings [21]. Studies of fibromyalgia and spinal cord injury also indicate that pain is reduced following five daily sessions of a-tDCS over the motor cortex [22,23].

A number of CLBP studies have examined the impact of tDCS over the motor cortex. Schabrun et al. [24] reported that a-tDCS over the motor cortex did not change motor cortex excitability in individuals with CLBP. Luedtke et al. [25] also reported that a-tDCS over the motor cortex had no therapeutic effect on CLBP. Hazime et al. [26] reported that a-tDCS over the motor cortex can induce a short-term and long-term analgesic effect in CLBP, but only when combined with peripheral electrical stimulation. A systematic review exploring the use of tDCS in CLBP concluded a Level A recommendation against the use of tDCS over the motor cortex, as it was shown to be ineffective in managing pain [27]. Subsequent research has turned its focus to investigated stimulation of other cortical areas involved in pain processing, such as the DLPFC [28,29,30].

The prefrontal cortex plays a key role in pain modulation synergistically through descending inhibition and cognitive-attentional mechanisms [31,32]. Functional imaging studies reveal that the DLPFC is involved in pain modulation [32,33]. Cao et al. [34] reported that stimulation of the DLPFC in healthy individuals can modulate motor cortex excitability via inhibitory and facilitatory connections. Imaging studies indicate that increased left-DLPFC activation is associated with reduced pain [32,33]. Further, repetitive transcranial magnetic stimulation (rTMS) over left-DLPFC is associated with reduced pain in those with migraine and fibromyalgia [28,29,30,35]. It has been suggested that such reductions in pain is associated with changes in motor cortex excitability [36]. Fierro et al. [36] reported that rTMS of the left-DLPFC in capsaicin-induced pain had an analgesic effect and reverted motor cortex excitability changes induced by the capsaicin pain stimulus. Vaseghi et al. [37] examined the impact of a-tDCS over left-DLPFC on motor cortex excitability in healthy individuals and found that a-tDCS increased the pressure-pain threshold and sensory threshold. These findings support the theory that the left-DLPFC and motor cortex play a key role in pain modulation and indicate that non-invasive brain stimulation to these areas might lead to changes that modify the pain experience. However, the impact of a-tDCS over left-DLPFC on motor cortex excitability in CLBP remains unclear and this in an area of increasing interest.

Individuals with chronic pain exhibit abnormal motor cortex excitability, and this abnormality is associated with increased pain levels [38,39]. A-tDCS over DLPFC restores normal inhibitory and excitatory systems and reduces pain levels in some forms of chronic pain [30,36]. It is unclear whether a-tDCS over DLPFC can modulate excitability in the motor cortex in people with CLBP and, if so, whether there is a corresponding reduction in pain. For the present study, the primary proposal was that 2-weekly, 1.5 mA a-tDCS over left DLPFC for 4 weeks would modulate excitability in people with CLBP, whereby participants in the a-tDCS group would demonstrate an increase in ICF and SICI, compared to the sham-tDCS (s-tDCS) group. It was secondarily proposed that the a-tDCS group would demonstrate a reduction in pain-related outcomes (pain intensity, disability, and pain catastrophising), compared to the s-tDCS group.

## 2. Materials and Methods

### 2.1. Participants

Participants took part in a 5-week randomised controlled trial approved by Curtin University Human Research Ethics Committee (approval number: HR17/2015; ACTRN12615000110583). Participants were recruited via convenience sampling between 2015–2018. All research was conducted in accordance with the Declaration of Helsinki and all participants provided written, informed consent. Study inclusion required a formal diagnosis of CLBP of at least 6 months duration by a qualified health professional (General Practitioner or Physiotherapist; A Transcranial Magnetic Stimulation (TMS) screening questionnaire [40] and a cognitive status assessment using the Telephone Interview for Cognitive Status –30 (score ≥ 18 for inclusion) was conducted to determine participant eligibility. Thirty-one participants met the inclusion criteria. Participants were randomised to the a-tDCS (anodal) or s-tDCS (sham) group. Of the 31 participants, four participants did not produce reliable MEPs and six participants had very high resting motor thresholds (rMT). Participants with very high thresholds were unable to participate in another aspect of the study about recruitment curves, and were subsequently excluded from the TMS measures analysed here. One participant was excluded due to ongoing muscle activation across multiple trials. One participant left the study prior to the intervention. Twelve participants (of the 31 who met inclusion criteria) were not included in the final TMS analysis presented here.

### 2.2. Measures

Demographic and pain-related information were collected via self-report questionnaire. Motor cortex excitability measures and pain-related measures were completed at baseline and immediately (~1 h) following the 4-week intervention.

#### 2.2.1. Motor Cortex Excitability Measures

EMG signals were recorded using Ag–AgCl surface electrodes placed over the belly and tendon of the left First Dorsal Interosseous (FDI) muscle. The EMG signal was sampled at 1000 Hz with a Power-1401 A/D board (Cambridge Electronic Design [CED], Cambridge, UK) and band-pass filtered at 5–500 Hz. The stimulation procedures were conducted using TMS. TMS was applied using a figure-of-eight coil (90 mm in diameter) connected to two Magstim 200 magnetic stimulators through a Bistim module (Magstim Company Limited, UK). The 10/20 International system for electrode placement was used to locate the motor area corresponding to the left FDI muscle [41]. The coil was positioned over the optimal location to produce a MEP in the contralateral FDI. The coil was placed at a 45-degree angle from the inter-hemispheric line (handle pointing towards the right), to stimulate current flow in a posterior to anterior direction. The FDI was chosen as the target area for stimulation as it has been reported that global alteration in cortical excitability can be reflected by responses to TMS of this muscle [6,42].

TMS stimulation intensity started at 30% (adjusted in 1% increments) until the rMT was established. rMT was defined as the lowest stimulation intensity that elicited MEPs ≥ 50 μV in at least 3 of 5 trials while the muscle was a rest [43,44]. The paired-pulse protocol developed by Kujirai et al. [45] was used to measure SICI and ICF. SICI and ICF were defined using a subthreshold conditioning pulse set to 80% of rMT, and a suprathreshold test pulse set at 120% of rMT [46]. The interstimulus interval was set to 3 ms and 10 ms for SICI and ICF, respectively. Fifteen trials were recorded at each interstimulus interval. Fifteen single unconditioned test pulses (at 120% rMT) were also recorded (set at 120% rMT). The order of administration was randomised. The fifteen trials for each interstimulus interval were averaged to attain a mean MEP amplitude. The mean MEP amplitude for SICI and ICF was normalised against the participant’s mean unconditioned pulse.

#### 2.2.2. Clinical Measures

Pain Intensity. The Short-Form McGill Pain Questionnaire (SF-MPQ) contains a 10 cm Visual Analogue Scale (VAS). The VAS was used to assess average pain intensity (scored from 0–10) in CLBP [47]. Participants indicated their level of pain by placing a mark on the line [48]. Higher scores are indicative of greater pain intensity. The VAS has high test–retest reliability in people with pain (0.97) [49].

Disability. The Roland-Morris Disability Questionnaire (RMDQ) assessed the level of self-reported disability [50]. The RMDQ assessed the impact of CLBP across multiple domains, including mobility, daily activities, sleeping, mood, and appetite. The 24 items are summed for a total score (higher scores are indicative of greater self-reported disability). The RMDQ has high internal consistency in people with CLBP (α = 0.93) [51].

Pain Catastrophising. The Pain Catastrophizing Scale (PCS) assessed the presence of pain catastrophising [52]. The PCS assesses rumination, magnification, and helplessness over 13 items, with. Higher scores indicating greater pain catastrophising. The PCS has high internal consistency in people with CLBP (α = 0.92) [53].

### 2.3. Brain Stimulation

Participants completed 8-sessions of tDCS stimulation over 4-weeks (2 sessions per week). tDCS was delivered using a constant current stimulator (Soterix®). Participants were randomly assigned (1:1 using block randomisation) to the anodal (a)-tDCS or sham (s)-tDCS group. The a-tDCS group received 20 min of constant 1.5 mA stimulation over left DLPFC every session. tDCS was delivered using two 35 cm^2^ sponge electrodes soaked in saline solution. According to the 10–20 international system for EEG electrode placement, the anode electrode was placed over F3 to stimulate the left DLPFC. The reference electrode was placed above the left eye, to ensure the current flowed through the prefrontal area. There was a ramp up period of 30 s at the beginning and 30 s ramp down at the end of the tDCS stimulation. Participants in the s-tDCS experienced the 30 s ramp up/down of tDCS (1.5 mA stimulation) at the commencement and end of the stimulation (20 min). The ramp up/down in the s-tDCS group at the beginning and end of the stimulation is designed to keep the participant blind to the stimulation group [18,54].

### 2.4. Statistical Analysis

R software (v4.1.2; R Foundation for Statistical Computing, Vienna, Austria) was used to conduct all analyses. All trials were visually inspected and peak to peak MEP amplitudes were manually marked. The EMG signal was screened for noise, artifacts, and voluntary contraction. Trials that were identified as obscuring the detection of the MEP amplitude were excluded from analysis. Trials were also excluded from further analysis if repeated with muscle activation was identified.

Motor Cortex Excitability Analysis. ICF, and SICI were analysed using linear mixed models using lme function, nlme package [55]. For the ICF and SICI models, tDCS group, time, and their interaction were included as fixed effects with the intercept of each participant modelled as a random effect.

Clinical Measures Analysis. Pain Intensity, RMDQ score, and Pain Catastrophising were analysed using linear mixed models. For all three models, tDCS group, time, and their interactions were included as fixed effects with the intercept of each participant modelled as a random effect. ID was included as a random effect.

Analyses were further supplemented with Bayesian linear mixed effects analyses using the rstanarm [56] and report [57] packages. In line with the Sequential Effect eXistence and sIgnificance Testing framework [58], the median of the posterior distribution, its 95% CI (Highest Density Interval), the probability of direction (pd), and the probability of significance are reported. Default weakly informative priors from the rstanarm package were used in analyses. The default Region of Practical Equivalence (ROPE) threshold, |0.05|, from the report package was used to assess the probability of significance. Values within this range are considered as practically equivalent to zero [58].

## 3. Results

Nineteen participants (from the sample of n = 30 from the overarching study) were included for analysis (see Figure 1). Participant demographic and pain-related information is provided in Table 1.

### 3.1. Test Pulse

Bootstrapped paired t-tests revealed no significant difference in pre and post test pulse MEP amplitude in the a-tDCS group and s-tDCS group, *p* = 0.527 and *p* = 0.708, respectively.

### 3.2. ICF

The linear mixed model revealed no significant main effect of Group (*p* = 0.428) or Time (*p* = 0.291). The interaction between Group and Time approached significance, *F*(1,17) = 4.07, *p* = 0.059. Further inspection of the plots suggests this interaction was driven by an increase in MEP amplitude in the a-tDCS group (see Figure 2A). At the suggestion of a reviewer, pain intensity was included in the model as a covariate. With the inclusion of pain intensity as a covariate, the Group and Time interaction became significant, *F*(1,13.63) = 6.02, *p* = 0.028.

A Bayesian linear mixed effects analysis compared pre- and post-intervention ICF MEP amplitude. The effect of tDCS group (anodal) had a probability of 96.79% [pd] of being positive (median = 0.47, 95% CI [−0.04, 0.96]), 96.16% of being significant, and 90.18% of being large (>0.15; see Figure 2B for differences).

### 3.3. SICI

There was no significant main effect of Group (*p* = 0.272) or Time (*p* = 0.139). The interaction between Group and Time approached significance, *F*(1,17) = 3.76, *p* = 0.069. Further inspection of the plots suggests this interaction was driven by an increase in MEP amplitude in the a-tDCS group (see Figure 3A). At the suggestion of the reviewer, pain intensity was included in the model as a covariate. The Group and Time interaction remained nonsignificant, *F*(1,16.00) = 4.28, *p* = 0.055.

A Bayesian linear mixed effects analysis compared pre and post intervention SICI MEP amplitude. The effect of tDCS group (anodal) had a probability of 96.60% [pd] of being positive (median = 0.20, 95% CI [−0.02, 0.43]), 95.73% of being significant, and 88.82% of being large (>0.07; see Figure 3B for differences).

### 3.4. Clinical Outcomes

Pain Intensity. The linear mixed model revealed no significant main effect of Group (*p* = 0.675). There was a significant main effect of Time, (*F*(1,16) = 7.74, *p* = 0.013). The interaction between Group and Time was not significant (*F*(1,16) = 3.13, *p* = 0.096). Although the interaction effect did not cross the statistical threshold, inspection of mean values in the plot suggests that the effect of pain before and after the intervention was driven by the active group, as pain scores in the sham group remained similar over time (see Figure 4A).

A Bayesian linear mixed effects analysis compared pre and post intervention pain intensity. The effect of tDCS group (anodal) had a probability of 94.96% [pd] of being negative (median = −1.38, 95% CI [−3.13, 0.32]), 93.92% of being significant, and 85.56% of being large (>0.51; see Table 2 and Figure 5A for differences).

Disability (RMDQ). The linear mixed model revealed no significant main effect of Group (*p* = 0.637) or Group and Time Interaction (*p* = 0.130). There was a significant main effect of Time, *F*(1,17) = 5.61, *p* = 0.03. Further inspection of the plots suggests this effect was driven by a decrease in RMDQ score in the a-tDCS group (see Figure 4B).

A Bayesian linear mixed effects analysis compared pre and post intervention disability. The effect of tDCS group (anodal) had a probability of 93.66% [pd] of being negative (median = −3.22, 95% CI [−7.53, 0.98]), 92.41% of being significant, and 82.14% of being large (>1.30; see Table 2 and Figure 5B).

Pain Catastrophising. The linear mixed model revealed no significant main effect of Group (*p* = 0.178) or Group and Time Interaction (*p* = 0.355). There was a significant main effect of Time, *F*(1,17) = 17.39, *p* < 0.001. Further inspection of the plots suggests pain catastrophising was reduced in both a-tDCS and s-tDCSgroup at post-intervention (see Figure 4C).

A Bayesian linear mixed effects analysis compared pre and post intervention catastrophising. The effect of tDCS group (anodal) had a probability of 81.87% [pd] of being positive (median = 3.18, 95% CI [−3.90, 10.59]), 78.89% of being significant, and 61.26% of being large (>2.11; see Table 2 and Figure 5C).

## 4. Discussion

The present study examined if 8 sessions (twice weekly) of 1.5 mA a-tDCS over left-DLPFC modulated motor cortical excitability and self-reported measures of pain and disability in those with CLBP. The interaction between tDCS group and time was not significant for both ICF and SICI, suggesting a-tDCS over left-DLPFC did not modulate motor cortex excitability. However, the interactions between tDCS group and time approached significance. Follow-up Bayesian analyses indicated that 8 sessions of a-tDCS over left-DLPFC may modulate ICF and SICI in those with CLBP. These interactions appear to be driven by an increase in ICF and SICI MEP amplitude in the a-tDCS group. For self-reported measures, tDCS did not impact upon pain intensity, disability, or catastrophising. However, tDCS impacted upon each measure independently. Changes in pain intensity and disability pre- and post-intervention were driven by a reduction in the a-tDCS group. These results are consistent with the theoretical framework that restoration of abnormal motor cortical excitability in CLBP may be associated with decreased pain and disability.

The present findings that a-tDCS over DLPFC may modulate motor cortex excitability extends on those of Vaseghi et al. [37]. Vaseghi et al. [37] used a-tDCS over DLPFC in motor excitability to explore the role of the DLPFC in pain in healthy adults. They reported increased MEP amplitude following 20-minute a-tDCS, which was evident both immediately following the stimulation and thirty minutes later. A-tDCS also increased the pressure-pain threshold and sensory threshold, suggesting that the DLPFC and the motor cortex play an important role in pain modulation. However, as Vaseghi et al. [37] examined motor cortical excitability using a single-pulse paradigm, the specific mechanisms underlying this cortico-pain relationship could not be determined. The present study extends on Vaseghi et al. [37] findings by specifically examining the impact of a-tDCS over DLPFC on ICF and SICI in CLBP.

ICF and SICI are indicators of glutamatergic (excitatory) and gamma-aminobutyric acid (GABA; inhibitory) neurotransmitters [62]. Glutamate and GABA are critical to maintaining and regulating many physiological functions and are reported to underlie altered excitability in varying chronic pain conditions [62,63]. The potential for tDCS over DLPFC to modulate ICF and SICI in chronic pain may be due to interconnections between the DLPFC and the periaqueductal gray area and motor cortex [64,65,66]. These areas play a key role in descending mechanisms that modulate spinal nociceptive activity [64,65,66]. A recent meta-analysis reported that while the balance of glutamate and GABA differed between chronic pain conditions, glutamate and GABA play a key role in the pathophysiology of pain processing and modulation [63]. This is supported in the present findings with increased ICF and SICI in the a-tDCS group corresponding with decreased pain and decreased perceived disability. Modulation of glutamatergic and GABAa via DLPFC stimulation may therefore play an important role in the relationship between corticospinal excitability and pain perception in CLBP.

In the present study, the reduction in pain intensity was greater for the a-tDCS group than for sham. To the best of our knowledge, no studies have investigated the use of a-tDCS over DLPFC to reduce pain in CLBP. Studies using another form of non-invasive brain stimulation (rTMS) yielded conflicting results. While the mechanisms of action for tDCS and rTMS may differ, both techniques may lead to similar changes in cortical activity and, as such, have a similar effect on chronic pain [21]. Freigang et al. [67] reported that, compared to motor cortex stimulation and sham, rTMS over left-DLPFC significantly reduced pain intensity in CLBP. In line with this, the present findings suggest that multi-session a-tDCS of left-DLPFC has an analgesic effect on CLBP. Perceived disability was also significantly reduced in the a-tDCS group compared to the s-tDCS group in the present study. Although no previous studies have examined the impact of left-DLPFC a-tDCS on perceived disability in CLBP, the present findings reflect those of Freigang et al. [67], who found that tDCS over DLPFC improved perceived health-related quality of life—physical functioning in those with CLBP. Taken together, these findings suggest that a-tDCS over DLPFC may modulate the cognitive and emotional appraisal of pain, which may manifest as a reduction in the pain experience for those with CLBP.

The present findings did not indicate that a-tDCS (compared to sham) was associated with reduced pain catastrophising. Both the anodal and the sham groups reported a reduction in pain catastrophising following tDCS, but this was not statistically significant. One potential explanation for the reduction in catastrophising in both groups may be the placebo effect. Engaging in the intervention may have had a positive impact on psychological health and wellbeing, such that catastrophising was reduced for both groups [68]. The pre-stimulation sensation (‘ramp-up’) in the sham group may have generated a treatment expectancy effect, whereby participants in the sham group believed they were receiving a-tDCS. However, the placebo effect does not account for the reduced pain and disability in the a-tDCS group only. It is also possible that participants experienced higher levels of pain-related anxiety before undertaking brain stimulation for the first time.

### Limitations

There are a number of limitations that must be acknowledged. Due to the nature of the over-arching project, participants were not required to complete or have useable TMS data to be included in the main project. This resulted in small, unequal group sizes in the present study. To accommodate the small sample size, Bayesian linear mixed models were conducted. Bayesian linear mixed models are better suited to handle small and unbalanced sample sizes, providing better estimates of the effects (see Hsieh and Maier [69], and McNeish [70], for review). Additionally, the interpretation of Bayesian linear mixed model analyses does not depend on the significance of *p*-values. Rather, Bayesian linear mixed models provide multiple alternative indices for *p*-values, and gives probabilities based on the measured data. As such, the results from the Bayesian linear mixed models were interpreted in the present study. While the findings from the Bayesian analyses suggest that a-tDCS over left-DLPFC may modulate motor cortex excitability and reduce pain intensity and disability in CLBP, it must be acknowledged that the interpretation of *p*-values of the primary analysis provide less clear-cut results. As such, the present findings should be interpreted with caution. Future research should include a larger sample size to not only establish the current findings, but allow for a more comprehensive examination of the relationships between motor cortex excitability, pain, disability, and pain catastrophising.

The use of analgesic medications (commonly prescribed for the management of CLBP) have been reported to influence cortical excitability. Benzodiazepines have been reported to significantly increase SICI and decrease ICF [71], while synthetic opioids (N-methyl-D-aspartate receptor agonists) have also been reported to decrease ICF [39]. Medication use was recorded in the present study, however, frequency of use and dosage was not documented. Future research should include a non-medicated sample to determine the influence of medication on the present findings, although given the chronic pain population sample, it may be difficult to find individuals who are not medicated.

The present study did not include measures of motor control. As motor cortical structures are involved in movement planning and execution [72], changes in motor cortex excitability and pain may be associated with changes in motor control in those with CLBP. Future research should include measures of motor control to examine if an increase in excitability and reduced pain intensity is associated with improved motor control in CLBP. This would require a longitudinal study of changes in excitability, pain, and motor control.

## 5. Conclusions

The present findings add to our understanding of the use of neuromodulation techniques to alleviate chronic pain. The present findings provide some preliminary evidence that repeated application of a-tDCS over left-DLPFC may modulate motor cortex excitability in CLBP. The present findings also provide preliminary evidence that a-tDCS over left-DLPFC may reduce pain and disability. While the findings of the present study should be interpreted with caution given the small sample size, the findings provide support for research to further investigate the potential of a-tDCS as a therapeutic tool in the management of CLBP. Future studies should examine if changes in motor cortex excitability underlie changes in CLBP symptoms and experience, and whether a-tDCS over left-DLPFC is a viable tool the management of CLBP.

## Figures and Tables

**Figure 1 brainsci-12-01654-f001:**
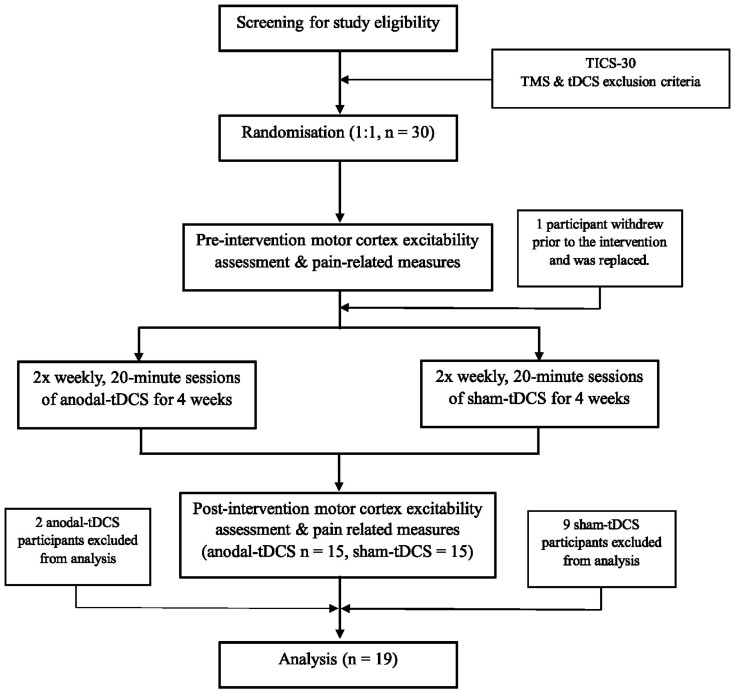
Flow diagram of the progress of the trial for anodal and sham transcranial direct current stimulation (tDCS) groups.

**Figure 2 brainsci-12-01654-f002:**
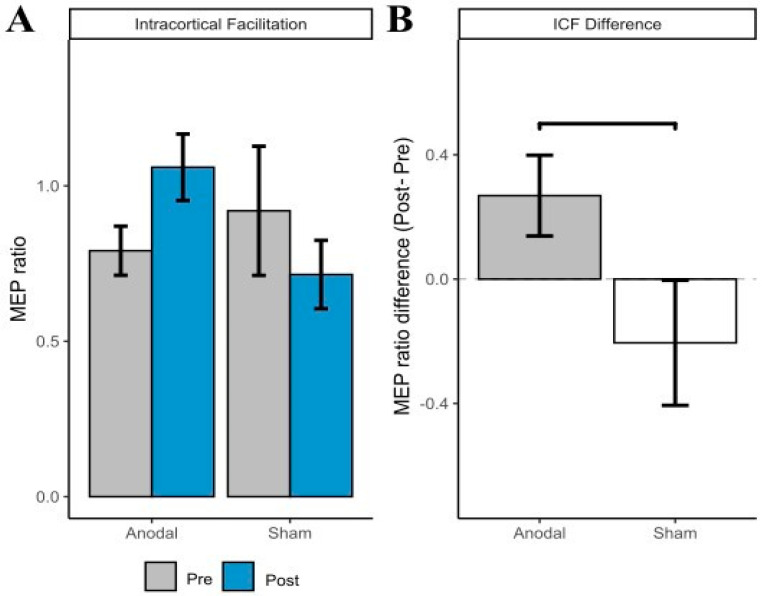
(**A**) Intracortical facilitation (ICF) motor evoked potential (MEP) amplitude (with standard error of the mean error bars) at pre- and post-intervention by group. (**B**) ICF difference score (Post ICF MEP amplitude—Pre ICF MEP amplitude; with standard error of the mean error bars) by group.

**Figure 3 brainsci-12-01654-f003:**
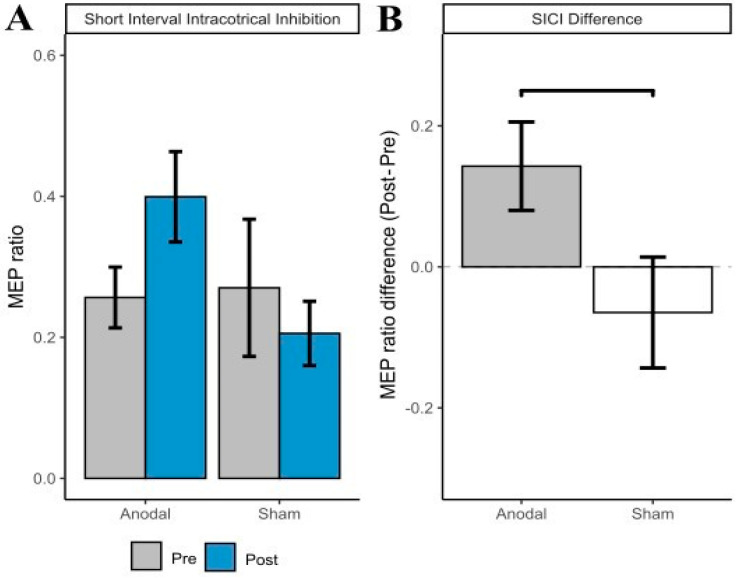
(**A**) Short interval intracortical inhibition (SICI) motor evoked potential (MEP) amplitude (with standard error of the mean error bars) at pre- and post-intervention by group. (**B**) SICI difference score (Post SICI MEP amplitude—Pre SICI MEP amplitude, with standard error of the mean error bars) by group.

**Figure 4 brainsci-12-01654-f004:**
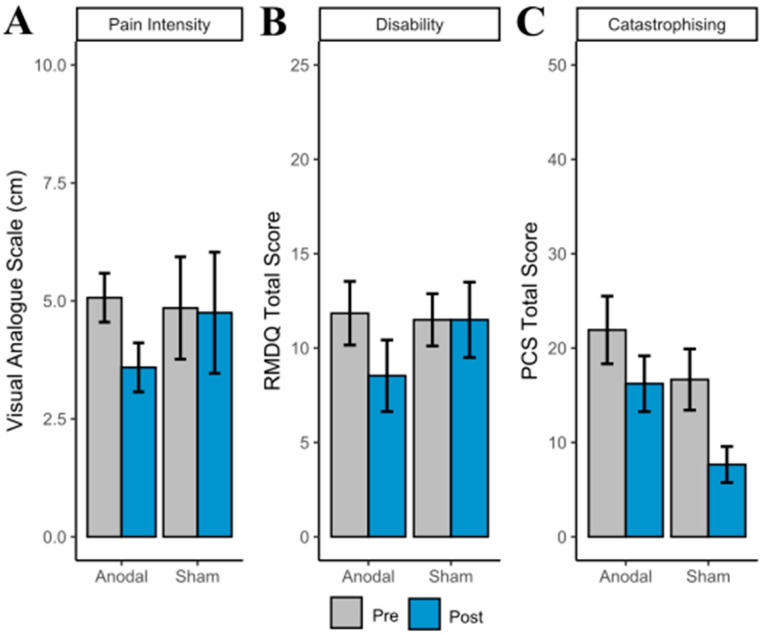
(**A**) Pain intensity, (**B**) Roland Morris Disability Questionnaire (RMDQ), and (**C**) Pain Catastrophising Scale (PCS) total scores (with standard error of the mean error bars) at pre- and post-intervention by group.

**Figure 5 brainsci-12-01654-f005:**
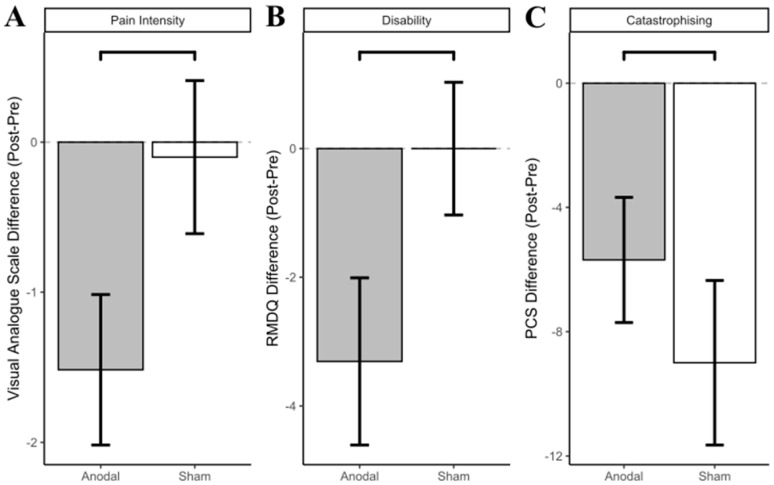
(**A**) Pain intensity, (**B**) Roland Morris Disability Questionnaire (RMDQ), and (**C**) Pain Catastrophising Scale (PCS) difference scores (Post score–pre score; with standard error of the mean error bars) by group.

**Table 1 brainsci-12-01654-t001:** Participant Demographics, CLBP Classification, and Treatment Engagement.

	Total (n = 19)	Males (n = 10)	Females (n = 9)	Anodal-tDCS	Sham-tDCS
Age	53.16 (14.80; 21–76)	57.80 (14.52)	48.00 (13.32)	49.69 (14.34)	60.67 (12.87)
Years of Education	12.65 (3.48)	12.55 (3.25)	12.77 (3.72)	12.53 (3.59)	12.92 (3.22)
Duration of Diagnosis (years)	13.38 (12.41)	17.93 (14.11)	8.33 (7.47)	12.31 (9.50)	15.71 (16.87)
Resting Motor Threshold	49.79 (8.75)	47.80 (5.90)	52.00 (11.07)	49.31 (9.12)	50.83 (8.59)
CLBP Classification					
Non-Specific	84%	80%	89%	85%	83%
Specific	16%	20%	11%	15%	17%
Percentage taking Pain Medication	74%	50%	100%	69%	83%
Anti-Inflammatory (Celebrex) ^a^	42%	40%	44%	54%	17%
Pain Killer (Tremadol) ^a^	53%	20%	89%	46%	67%
Benzodiazepine (Valium) ^a^	21%	10%	33%	23%	17%
Anti-Depressants (Endep) ^a^	5%	10%	-	8%	-
Engaging in Physiotherapy	47%	30%	67%	54%	33%
Past Surgery	21%	20%	22%	77%	17%
Other Pain Management	79%	80%	78%	15%	67%
Depression and Anxiety Disorder	16%	10%	22%	15%	17%
Anti-Anxiety Medication ^a^	33%	-	50%	50%	-

Note. CLBP Classification = classification of chronic lower back pain based on Koes et al. [59], Non-Specific = no radiographical support of injury at time of participation, Specific = Radiographical evidence of injury, Other = Acupuncture, Chiropractor, Massage. ^a^ = Percentage based on individuals taking pain management medication.

**Table 2 brainsci-12-01654-t002:** Mean and standard deviation for Pain intensity, disability, and pain catastrophising pre- and post-intervention.

	Anodal-tDCS		Sham-tDCS	
	Pre-Intervention	Post-Intervention	Pre-Intervention	Post-Intervention
VAS	5.07 (1.87)	3.59 (1.72) ^a^	4.85 (2.66)	4.75 (3.14)
RMDQ	11.85 (6.08)	8.54 (6.84) ^b^	11.50 (3.39)	11.50 (4.89)
PCS	21.92 (12.93)	16.23 (10.67) ^c^	16.67 (7.94)	7.67 (4.68)

Note. VAS = Visual Analogue Scale (pain intensity), RMDQ = Roland Morris Disability Questionnaire (disability), PCS = Pain Catastrophising Scale (catastrophising). ^a^ = borderline minimal detectable change (<1.5; [60]), ^b^ = reached minimal detectable change (≥2.5) [60], ^c^ = did not reach minimally clinically important difference (<6.71) [61].

## Data Availability

Data is available on reasonable request from the corresponding author.

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
