# Peer review of "Anodal-TDCS over Left-DLPFC Modulates Motor Cortex Excitability in Chronic Lower Back Pain"

_brainsci, 2022, doi:10.3390/brainsci12121654_

Round 1

Reviewer 1 Report

Overall the manuscript is well written and the experiment and results are clearly described.  I thought the interpretations were generous given that the primary analyses typically failed to show significant effects; whereas the secondary analyses (Bayesian) provided significant effects.  I think the author should discuss how this was the case and why the reader should interpret the later rather than the former set of analyses.  Limitations were appropriately acknowledged; however, it appeared that ~38% of the total participants data (n=31) was not included in the analysis.  I think the author should acknowledge the weakness of their approach for studying cortical excitability if nearly 40% of participants data can not be analyzed.  The problem with tDCS is variability within and across subjects.  How specifically then does this study support the use of tDCS for pain intervention if it can not be used reliably to modulate cortical excitability in 38% of subjects.  It seems that global statements about whether or not tDCS over the DLPFC works to reduce cortical excitability and pain ratings in CLBP are not warranted at this stage.  Rather, there may be a subset of patients for whom this approach works and effort should be placed in understanding the factors that make a positive response or the use of this approach more likely.

Author Response

Note: Please see the attachment for responses to all reviewers. Responses to Reviewer 1 below;

Comment 1: Overall the manuscript is well written and the experiment and results are clearly described.  I thought the interpretations were generous given that the primary analyses typically failed to show significant effects; whereas the secondary analyses (Bayesian) provided significant effects.  I think the author should discuss how this was the case and why the reader should interpret the later rather than the former set of analyses. 

Response: We thank the reviewer for the above comment. The decision to include and interpret the Bayesian analysis over the primary analysis is because Bayesian linear mixed model analysis is more appropriate to deal with smaller sample sizes (see Hsieh & Maier, 2009 for review) and the interpretation of results doesn’t depend on the significance of p-values. P-values are partly determined by data that has not been measured, whereas Bayesian analyses uses probabilities of the measured data (i.e., what is the probability of the hypothesis given the measured data). There is growing evidence that the reliance on p-values, particularly in clinical trials, leads to incorrectly interpretating that an intervention had no effect, simply because the p value was categorised as non-significant (see Amrhein et al., 2019 for review). The Bayesian linear mixed model provides alternative indices for the p-value. In our study, we have provided several alternative indices, namely, Region of Practical Equivalence (ROPE), probability of direction (pd), and median of the posterior distribution, and its 95% CI (Highest Density Interval). We do acknowledge that both frequentist and Bayesian methods have advantages and disadvantages, however, the decision to interpret the Bayesian linear mixed models is informed by its appropriateness for small, un-balanced samples, provides multiple alternative indices for the p-value, and provides probabilities based on the measured data.

The following information has been included in the manuscript to reflect the decision to interpret the Bayesian analyses; “There are a number of limitations that must be acknowledged. Due to the nature of the over-arching project, participants were not required to complete or have useable TMS data to be included in the main project. This resulted in small, unequal group sizes in the present study. To accommodate the small sample size, Bayesian linear mixed models were conducted. Bayesian linear mixed models are better suited to handle small and unbalanced sample sizes, providing better estimates of the effects (see Hsieh & Maier, 2009 and McNeish 2016 for review). Additionally, the interpretation of Bayesian linear mixed model analyses does not depend on the significance of p-values. Rather, Bayesian linear mixed models provide multiple alternative indices for p-values, and give probabilities based on the measured data. As such, the results from the Bayesian linear mixed models were interpreted in the present study. While the findings from the Bayesian analyses suggest that a-tDCS over left-DLPFC may modulate motor cortex excitability and reduce pain intensity and disability in CLBP, it must be acknowledged that the interpretation of p-values of the primary analysis provided less clear-cut results. As such, the present findings should be interpreted with caution.” (see page 11,line 372-386)

Amrhein, V., Trafimow, D., & Greenland, S. (2019) Inferential statistics as descriptive statistics: There is no replication crisis if we don’t expect replication, The American Statistician, 73:sup1, 262-270, https://doi.org/10.1080/00031305.2018.1543137

Hsieh, C-A., & Maier, K.S. (2009) A preliminary Bayesian analysis of incomplete longitudinal data from a small sample: methodological advances in an international comparative study of educational inequality. International Journal of Research & Method in Education, 32(1), 103-125, https://doi.org/10.1080/17437270902749353

McNeish, D. (2016) On using Bayesian methods to address small sample problems. Structural Equation Modeling: A Multidisciplinary Journal, 23(5), 750-773, https://doi.org/10.1080/10705511.2016.1186549

Comment 2: Limitations were appropriately acknowledged; however, it appeared that ~38% of the total participants data (n=31) was not included in the analysis.  I think the author should acknowledge the weakness of their approach for studying cortical excitability if nearly 40% of participants data cannot be analyzed.  The problem with tDCS is variability within and across subjects.  How specifically then does this study support the use of tDCS for pain intervention if it cannot be used reliably to modulate cortical excitability in 38% of subjects. 

Response: We apologise for not making the reason behind exclusion clearer. Due to the overarching nature of the project, it was not a requirement that all participants produce reliable TMS data to be retained in the overarching project. At the baseline of the overarching project, 21 participants (of n = 31) produced reliable TMS data. Of these 21 participants, 19 participants had reliable TMS data at baseline and post-intervention. As such, > 90% of the participants with reliable TMS data at baseline, also had reliable data at post-intervention.

Comment 3: It seems that global statements about whether or not tDCS over the DLPFC works to reduce cortical excitability and pain ratings in CLBP are not warranted at this stage.  Rather, there may be a subset of patients for whom this approach works and effort should be placed in understanding the factors that make a positive response or the use of this approach more likely.

Response: We apologise for overstating the conclusions of the present study. In the conclusion, we have highlighted that the results are preliminary and should be interpreted with caution. We have also suggested that these findings provide early support that research needs to investigate the potential for tDCS to be used as a therapeutic tool.  The following has been included in the manuscript; “The present findings provide some preliminary evidence that repeated application of a-tDCS over left-DLPFC may modulate motor cortex excitability in CLBP. The present findings also provide preliminary evidence that a-tDCS over left-DLPFC may reduce pain and disability. While the findings of the present study should be interpreted with caution given the small sample size, the findings provide support for research to further investigate the potential of a-tDCS as a therapeutic tool in the management of CLBP” (see page 11, line 409-417).

Reviewer 2 Report

Overview and general recommendation:

This study tested a novel hypothesis on whether anodal-TDCS over left DLPFC modulates motor cortex excitability in population with chronic lower back pain (CLBP).  The authors conducted a between-subject design with participants receiving either a-tDCS or sham. The authors clearly made two specific predictions on the experiment: (1) a-tDCS group will show an increase in ICF and SICI (indicators of motor cortex excitability), compared with sham; (2) a-tDCS group will show a reduction in three pain related outcomes, compared with sham.

The study has been well planned and conducted, with a proper sham control condition. The statistical analyses were properly conducted and reported. I also found the paper to be overall well written with many important technical details clearly organized. However, my main concern is the small sample size of the study, especially considering this is a between-subject design. The fact that many of the participants are on their medications also added complexity to each group of participants. Most of the statistical analysis results yielded very large variance, which prevents the authors from reaching statistical significance and arriving at a convincing conclusion.

Major comments:

1.     As I mentioned above, this study can only reach a convincing conclusion with a larger sample size.

2.     The age of the participation population has a large variance (Table 1). It will be helpful to indicate the range of age.

Minor comments:

1.     Figure 2,3,4,5. These figures use gray and blue colors for pre and post test, respectively, as well as for anodal and sham, respectively. This might cause some confusion when reading the figures. I suggest the authors use consistent color scale, and add use colors of different intensity if needed.

2.     The authors should use either uppercase or lowercase to denote the figure panels, not to mix them in figures and texts.

3.     The title should not omit “left” in left DLPFC to properly express the key information

4.     Lack of clarity in writing: line 41-42, it confused me how GABAA and glutamate neurotransmitters are relevant to SICI or ICF. This relationship was not clear until line 302-304 in the discussion section.

Author Response

Note: Please see the attachment for responses to all reviewers. Responses to Reviewer 2 provided below;

This study tested a novel hypothesis on whether anodal-TDCS over left DLPFC modulates motor cortex excitability in population with chronic lower back pain (CLBP).  The authors conducted a between-subject design with participants receiving either a-tDCS or sham. The authors clearly made two specific predictions on the experiment: (1) a-tDCS group will show an increase in ICF and SICI (indicators of motor cortex excitability), compared with sham; (2) a-tDCS group will show a reduction in three pain related outcomes, compared with sham.

The study has been well planned and conducted, with a proper sham control condition. The statistical analyses were properly conducted and reported. I also found the paper to be overall well written with many important technical details clearly organized. However, my main concern is the small sample size of the study, especially considering this is a between-subject design. The fact that many of the participants are on their medications also added complexity to each group of participants. Most of the statistical analysis results yielded very large variance, which prevents the authors from reaching statistical significance and arriving at a convincing conclusion.

Comment 1: As I mentioned above, this study can only reach a convincing conclusion with a larger sample size.

Response: We apologise for over-stating the findings of the study. We have amended the conclusion of the manuscript to reflect that these findings are preliminary in nature and that caution must be used in interpreting the results (please see response to Reviewer 1, comment 3).

Comment 2: The age of the participation population has a large variance (Table 1). It will be helpful to indicate the range of age.

Response: The age range of the participants has been included in Table 1.

Comment 3: Figure 2,3,4,5. These figures use gray and blue colors for pre and post test, respectively, as well as for anodal and sham, respectively. This might cause some confusion when reading the figures. I suggest the authors use consistent color scale, and add use colors of different intensity if needed.

Response: We apologise for the oversight. We have amended the figures such that the differentiation between pre and post, and anodal and sham, are clear.

Comment 4: The authors should use either uppercase or lowercase to denote the figure panels, not to mix them in figures and texts.

Response: We apologise for the oversight. This has been amended in the figures and in text.

Comment 5: The title should not omit “left” in left DLPFC to properly express the key information

Response: “left” has now been included in the title.

Comment 6: Lack of clarity in writing: line 41-42, it confused me how GABAA and glutamate neurotransmitters are relevant to SICI or ICF. This relationship was not clear until line 302-304 in the discussion section.

Response: The following information has been included in the manuscript to provide clarity about the relationship between GABAA and glutamate, and SICI and ICF; “Research in experimentally induced and chronic pain have suggested that short interval intracortical inhibition (SICI) and intracortical facilitation (ICF) may be the key mechanisms associated with the maintenance of pain [12], whereby an imbalance between inhibition (GABAA; indicated by SICI) and excitation (glutamate; indicated by ICF) may be associated with increased pain intensity” (see page 1, line 38-43).

Reviewer 3 Report

The purpose of this clinical trial entitled "Anodal-TDCS over DLPFC modulates motor cortex excitability in chronic lower back pain" is to evaluate the effectiveness of anodal tDCS over DLPFC in chronic low back pain, taking motor cortex excitability and other clinical variables as outcomes. This is a pertinent study, with significant interest in the study of chronic pain. In the opinion of this reviewer, the following suggestions and recommendations should be taken into account to improve the quality of the manuscript and be considered for subsequent publication:

TITLE.

Report the type of study performed.

ABSTRACT.

Please exercise caution when reporting positive results (e.g., pain) that are not significant. Likewise, it would be recommendable to review pain intensity, disability, and pain catastrophizing as secondary variables.

INTRODUCTION.

The introduction is well prepared, with a correct exposition of the problem and the existing gap in the topic. The study is well justified.

P2Ln90. Better to change “2x weekly” to “2-weekly sessions”.

P2Ln90-95. These lines are considered as the primary and secondary hypothesis of the study. Please write the objectives of the study explicitly below.

MATERIAL AND METHODS

P2Ln96. Report the type of study performed.

P2Ln97. Where were the participants recruited from? What type of sampling was done? What was the time period in which the study was conducted? Was the protocol previously registered in any database?.

P3Ln98. Was a sample size calculation performed prior to the study? If this were not the case, the statistical power of the study might not be correct. Discuss later.

P3Ln101. List more clearly the inclusion and exclusion criteria. As written it creates some confusion. Include in this paragraph the possible exclusion criteria related to the application of TMS or tDCS.

P3Ln107-114. The data related to the number of participants recruited or finally included must be reflected in the Results section. Table 1 also belongs to that section. The causes of drop-outs should not be written in Material and methods either.

P3Ln107. According to the interpretation of this reviewer, the aspect that differentiates both groups is the application or not of tDCS current, and not the placement of the electrodes (anode). Thus, the most correct would be to name the groups as active anodal-tDCS (active a-tDCS) and sham anodal-tDCS (sham t-DCS). Apply equally to the abstract.

P3Ln114. A subheading titled “Randomization and blinding” should be included detailing how the randomization of the subjects was performed. Likewise, it must be reflected which researchers remained blind to the assignment of the subjects (evaluator, therapist...) and if the subjects were also unaware of the group to which they belonged.

P4Figure1. This figure has to be moved to the Results section.

P4Ln123. Please list more clearly the study variables and the time when the data was collected. Was the evaluator the same investigator who applied the treatments? Was the evaluator aware of the assignment of subjects to these groups?

P4Ln127. It would be advisable to divide the variables into primary and secondary for a better understanding of the text.

P4Ln127. Please define here the concepts of SICI and ICF, and how they are to be interpreted with respect to the assessment of motor cortex excitability.

P5Ln154. Why is this SF-MPQ used if only the VAS scale was used? It would have been better to use a VAS scale directly.

P5Ln169. There is some confusion for the reader in this subheading. Please write two paragraphs: one for active a-tDCS and one for sham a-tDCS.

Do you mean to refer to the ipsilateral Fp2 position as above the eye region?

Is it a montage resulting from a decision by the researchers or based on previous studies? Please, there are missing references throughout this section.

In the active a-tDCS setup, was there a 30 second ramp up, 19 minute current at 1.5mA, and a 30 second ramp down? What do you mean by “designed to mask the sham condition”?

In the paragraph corresponding to the sham a-tDCS, it should be reflected if the same methodology was used as in the active a-tDCS (electrostimulator, electrode placement, etc). Is it the FISSFO protocol? Since current is applied for 30 seconds at the start and end, the phrase “current ceased after 30 seconds” should be removed. Please provide a reference to a previous trial that has used this setup for the sham condition.

Are there any studies that show that this short 30 second stimulation does not produce brain stimulation? Provide reference.

RESULTS

P6Ln205. Were there significant differences between baseline groups? Were there significant differences in dropouts from each group? Please discuss the possible reason and its implication on the results.

P8Figure4. Please, keep the same order of both groups in all your figures. In this figure, the sham group appears on the left and the active group on the right, unlike in the rest of the figures.

DISCUSSION

P8Ln280. Include in this line the number of sessions carried out. Explicitly reflect in these first lines that there were no significant differences, or what is the same, that the active a-tDCS was not superior to the sham a-tDCS.

P9Ln284. The results should be interpreted with caution given the lack of significant results, the high number of dropouts, and the small sample size of the study.

CONCLUSIONS

P10Ln368-370. This statement should be removed, as it should not be interpreted as a conclusion that it is the first study to evaluate ICF and SICI using tDCS on DLPFC.

State conclusions with caution due to the lack of significant results and the low statistical power of the study.

Author Response

Note: Please see the attachment for responses to all reviewers. Responses to Reviewer 3 provided below;

The purpose of this clinical trial entitled "Anodal-TDCS over DLPFC modulates motor cortex excitability in chronic lower back pain" is to evaluate the effectiveness of anodal tDCS over DLPFC in chronic low back pain, taking motor cortex excitability and other clinical variables as outcomes. This is a pertinent study, with significant interest in the study of chronic pain. In the opinion of this reviewer, the following suggestions and recommendations should be taken into account to improve the quality of the manuscript and be considered for subsequent publication.

Comment 1: Please exercise caution when reporting positive results (e.g., pain) that are not significant. Likewise, it would be recommendable to review pain intensity, disability, and pain catastrophizing as secondary variables

Response: We thank the reviewer for this comment and have re-run the analysis including pain intensity in the model of ICF and SICI, with pain intensity as a continuous covariate (disability and pain catastrophising were not included in the model as they were highly correlated with pain intensity). The inclusion of pain intensity to the Group and Time interaction for SICI, although approaching, remained non-significant (p = .055). However, the inclusion of pain intensity as a continuous covariate in the ICF model caused the Group and Time interaction for ICF to become significant (p = .028). We have included these findings in the results section following the primary analysis (i.e., without pain intensity as a covariate; see page 6, line 234-237 and page 7, line 250-252). Given the small sample size, however, we are still suggesting this result be interpreted with caution.

Comment 2: P2Ln90. Better to change “2x weekly” to “2-weekly sessions.

Response: This has been amended.

Comment 3: P2Ln90-95. These lines are considered as the primary and secondary hypothesis of the study. Please write the objectives of the study explicitly below.

Response: We have now indicated the impact of a-tDCS on motor cortex excitability as the primary hypothesis, and the impact of s-tDCS on pain-related outcomes as the secondary hypothesis 

Comment 4: P2Ln96. Report the type of study performed.

Response: This has been included in the manuscript (see page 3, line 99).

Comment 6: P2Ln97. Where were the participants recruited from? What type of sampling was done? What was the time period in which the study was conducted? Was the protocol previously registered in any database?.

Response: We apologise for not including this information in the original manuscript. The following information has been included; “Participants were recruited via convenience sampling between 2015-2018.” Additionally, the study was registered with the Australian New Zealand Clinical Trials Registry. The registration number was included in the manuscript (ACTRN12615000110583; see page 3, line 101-102).

Comment 7: P3Ln98. Was a sample size calculation performed prior to the study? If this were not the case, the statistical power of the study might not be correct. Discuss later.

Response: A sample size calculation was conducted for the overarching study. We have discussed the small sample in the present study in the limitations; “Due to the nature of the over-arching project, participants were not required to complete or have useable TMS data to be included in the main project. This resulted in small, unequal group sizes in the present study” (see page 11, line 372-374), and in the conclusion; “the findings of the present study should be interpreted with caution given the small sample size” (see page 11, line 412-413).

Comment 8: P3Ln101. List more clearly the inclusion and exclusion criteria. As written it creates some confusion. Include in this paragraph the possible exclusion criteria related to the application of TMS or tDCS.

Response: The exclusion criteria for TMS and/or tDCS is well established. The article referenced in the present paper, Rossi et al. (2009), provides the safety and ethical guidelines for the use of TMS in practice and proposed a standard screening questionnaire on behalf of the International Federation of Clinical Neurophysiology. As such, we do not feel that it is required to list the exclusion criteria in the paper. Additionally, there is a significant overlap between the exclusion criteria for TMS and tDCS. TMS has an extensive criterion, and as such this was the criteria applied for suitability for both TMS and tDCS. This is also common in TMS and tDCS research. However, if the reviewer and editor strongly disagree, we agree to include this information in any subsequent revisions.

Comment 9: P3Ln107-114. The data related to the number of participants recruited or finally included must be reflected in the Results section. Table 1 also belongs to that section. The causes of drop-outs should not be written in Material and methods either.

Response: We thank the reviewer for their comment, however, we feel that the method section is the appropriate section for this information. Table 1 provides the descriptive characteristics of the population included and we feel the description of participants is most appropriate in the methods section. Additionally, as this study forms part of an overarching study, it was not a requirement that participants have reliable TMS data to be included in the overarching project. Please see response to reviewer 1, comment 2 regarding the participants included in the study.

Comment 10: P3Ln107. According to the interpretation of this reviewer, the aspect that differentiates both groups is the application or not of tDCS current, and not the placement of the electrodes (anode). Thus, the most correct would be to name the groups as active anodal-tDCS (active a-tDCS) and sham anodal-tDCS (sham t-DCS). Apply equally to the abstract.

Response to reviewer: We apologise if the difference between anodal-tDCS and sham-tDCS was not clear. Anodal-tDCS is considered to be active stimulation and is typically reported as anodal-tDCS in research. Sham-tDCS as a control condition is also typically referred to as Sham-tDCS. As such, we request to keep the current terminology as this aligns with research in this field. If the reviewer and editor strongly disagree with this terminology, we would be willing to change this in subsequent revisions.

Comment 11: P3Ln114. A subheading titled “Randomization and blinding” should be included detailing how the randomization of the subjects was performed. Likewise, it must be reflected which researchers remained blind to the assignment of the subjects (evaluator, therapist...) and if the subjects were also unaware of the group to which they belonged.

Response: Please see our response below (comment 19) discussing the meaning behind “designed to mask the sham condition”. Participants were unaware of the group to which they belonged. A separate member of the study team conducted the randomisation to the researchers that conducted the TMS and tDCS procedures. Due to the nature of the equipment (requires manual selection of anodal or sham stimulation) the study is single-blind in nature. The following information has been included in the manuscript; “Participants were randomly assigned (1:1 using block randomisation) to the anodal (a)-tDCS or sham (s)-tDCS group” (see page 5, 185-187).

Comment 12: P4Figure1. This figure has to be moved to the Results section.

Response: As per the comment above re Table 1 (comment 9), we feel that Figure 1 is best placed in the method section, as it provides an overview of the study procedure. As such we request to keep Figure 1 as part of the methods section. If the reviewer and editor strongly disagree with this, we would be willing to change this in subsequent revisions.

Comment 13: P4Ln123. Please list more clearly the study variables and the time when the data was collected. Was the evaluator the same investigator who applied the treatments? Was the evaluator aware of the assignment of subjects to these groups?

Response: Please see response to comment 11.

Comment 14: P4Ln127. Please define here the concepts of SICI and ICF, and how they are to be interpreted with respect to the assessment of motor cortex excitability.

Response: The following information has been included in the introduction of the manuscript to provide clarity about the relationship between GABAA and glutamate, and SICI and ICF; “Research in experimentally induced and chronic pain have suggested that short interval intracortical inhibition (SICI) and intracortical facilitation (ICF) may be the key mechanisms associated with the maintenance of pain [12], whereby an imbalance between inhibition (GABAA; indicated by SICI) and excitation (glutamate; indicated by ICF) may be associated with increased pain intensity” (see page 1, line 38-43)

Comment 15: P5Ln154. Why is this SF-MPQ used if only the VAS scale was used? It would have been better to use a VAS scale directly.

Response: The inclusion of the SF-MPQ is due to the overarching nature of the study, in which more detailed descriptive information in relation to type of pain sensations was collected. Given the small sample size for this aspect of the study, only pain intensity was included.

Comment 16: P5Ln169. There is some confusion for the reader in this subheading. Please write two paragraphs: one for active a-tDCS and one for sham a-tDCS.

Response: We apologise that we did not make this section clearer. This section has been edited for clarity.

Comment 17: Do you mean to refer to the ipsilateral Fp2 position as above the eye region?

Response: For the location of the reference electrode, we did not adhere to the 10/20 EEG placement, such that the electrode was not placed according to a specific EEG placement. This is consistent with unilateral and monopolar montages, where the reference electrode may be placed on a part of the body away from the head (Nasseri et al., 2015).

Comment 18: Is it a montage resulting from a decision by the researchers or based on previous studies? Please, there are missing references throughout this section.

The montage in the current study has been utilised in a tDCS study in Parkinson’s disease (Lawrence et al., 2018). Similar montages, i.e., with similar distances between the anode and reference electrode (as per the 10-20 system) and with 35cm2 sponges, have also been reported (Hazari et al., 2016; Kincses et al., 2004; Narayanaswamy et al., 2015).

Lawrence, B. J., Gasson, N., Johnson, A. R., Booth, L., & Loftus, A. M. (2018). Cognitive Training and Transcranial Direct Current Stimulation for Mild Cognitive Impairment in Parkinson's Disease: A Randomized Controlled Trial. Parkinson's disease, 4318475. https://doi.org/10.1155/2018/4318475

Hazari, N., Narayanaswamy, J.C., Chhabra, H., Bose, A., Venkatasubramanian, G., & Reddy, J.Y.C. (2016). Response to Transcranial Direct Current Stimulation in a Case of Episodic Obsessive Compulsive Disorder. The Journal of ECT, 32, 144-146 https://doi.org/10.1097/YCT.0000000000000309

Kincses, T. Z., Antal, A., Nitsche, M. A., Bártfai, O., & Paulus, W. (2004). Facilitation of probabilistic classification learning by transcranial direct current stimulation of the prefrontal cortex in the human. Neuropsychologia, 42(1), 113–117. https://doi.org/10.1016/s0028-3932(03)00124-6

Narayanaswamy, J. C., Jose, D., Chhabra, H., Agarwal, S. M., Shrinivasa, B., Hegde, A., Bose, A., Kalmady, S. V., Venkatasubramanian, G., & Reddy, Y. C. (2015). Successful Application of Add-on Transcranial Direct Current Stimulation (tDCS) for Treatment of SSRI Resistant OCD. Brain stimulation, 8(3), 655–657. https://doi.org/10.1016/j.brs.2014.12.003

Comment 19: In the active a-tDCS setup, was there a 30 second ramp up, 19 minute current at 1.5mA, and a 30 second ramp down? What do you mean by “designed to mask the sham condition”?

Response: We apologise that this statement was not clear. This section has been amended as follows; “There was a ramp up period of 30 seconds at the beginning and 30 seconds ramp down at the end of the tDCS stimulation. Participants in the s-tDCS experienced the 30 second ramp up/down of tDCS (1.5 mA stimulation) at the commencement and end of the stimulation (20 minutes). The ramp up/down in the s-tDCS group at the beginning and end of the stimulation is designed to keep the participant blind to the stimulation group” (see page 35, line 193-198).

Comment 20: In the paragraph corresponding to the sham a-tDCS, it should be reflected if the same methodology was used as in the active a-tDCS (electrostimulator, electrode placement, etc). Is it the FISSFO protocol? Since current is applied for 30 seconds at the start and end, the phrase “current ceased after 30 seconds” should be removed. Please provide a reference to a previous trial that has used this setup for the sham condition. Are there any studies that show that this short 30 second stimulation does not produce brain stimulation? Provide reference.

Response: We apologise if this was not clear. The method of ramp up/down is the most common method of sham control in tDCS research (Ambrus et al., 2012). The use of the ramp up/down method is to replicate the itching/tingling sensations that are typically reported within the first few moments of the stimulator being turned on. This brief stimulation period does not change cortical excitability (Nitsche et al., 2008).

Ambrus, G. G., Al-Moyed, H., Chaieb, L., Sarp, L., Antal, A., & Paulus, W. (2012). The fade-in--short stimulation--fade out approach to sham tDCS--reliable at 1 mA for naïve and experienced subjects, but not investigators. Brain stimulation, 5(4), 499–504. https://doi.org/10.1016/j.brs.2011.12.001

Nitsche, M. A., Cohen, L. G., Wassermann, E. M., Priori, A., Lang, N., Antal, A., Paulus, W., Hummel, F., Boggio, P. S., Fregni, F., & Pascual-Leone, A. (2008). Transcranial direct current stimulation: State of the art 2008. Brain stimulation, 1(3), 206–223. https://doi.org/10.1016/j.brs.2008.06.004

Comment 21: P6Ln205. Were there significant differences between baseline groups? Were there significant differences in dropouts from each group? Please discuss the possible reason and its implication on the results.

Response: The two groups did not significantly differ on duration of pain, resting motor threshold, or use of medication at baseline. The two groups did not significantly differ on pain intensity, disability, or pain catastrophising at baseline. There was a significant difference in age between the two groups. This significance appeared to be driven by the inclusion of 1 participant who was significantly younger than the group average (21 years of age). To determine if age needed to be included as a covariate, we compared the two models (with and without age). The model including age was not a significantly better fit than the original model, and as such the original model (without age) was retained for analysis. Please see Reviewer 1, comment 2 in regards to participant drop out.  

Comment 22: P8Figure4. Please, keep the same order of both groups in all your figures. In this figure, the sham group appears on the left and the active group on the right, unlike in the rest of the figures.

Response: We apologise for this oversight. The figures have been amended such that the Anodal group always appears first in all figures.

Comment 23: P8Ln280. Include in this line the number of sessions carried out. Explicitly reflect in these first lines that there were no significant differences, or what is the same, that the active a-tDCS was not superior to the sham a-tDCS.

Response: This has been amended to read; “The present study examined if 8 sessions (twice weekly) of 1.5mA a-tDCS over left-DLPFC modulated motor cortical excitability and self-reported measures of pain and disability in those with CLBP. The interaction between tDCS group and time was not significant for both ICF and SICI, suggesting a-tDCS over left-DLPFC did not modulate motor cortex excitability” (see page 9, line 305-309).

Comment 24: P9Ln284. The results should be interpreted with caution given the lack of significant results, the high number of dropouts, and the small sample size of the study.

Please see response to reviewer 1, comment 1 & 3, in regard to toning down the conclusion of the study and providing more detail to the limitations. 

Comment 25: P10Ln368-370. This statement should be removed, as it should not be interpreted as a conclusion that it is the first study to evaluate ICF and SICI using tDCS on DLPFC.

Response: This statement has now been removed.

Round 2

Reviewer 2 Report

My comments have been well addressed by the authors. I thank the authors for spending their time answering them. 

Author Response

Thank you for your time.

Reviewer 3 Report

The manuscript has been meticulously revised and improved by the authors. Its quality is appropriate in the journal. Even so, I would like you to consider the following last suggestions:

Figure 1 and table 1 must be included in the Results section, as indicated by the CONSORT statement guidelines for RCTs (https://www.consort-statement.org/).

Regarding comments 19 and 20, this reviewer fully agrees with the authors. Please, add in the text the references provided in the cover letter.

Author Response

Thank you for your time reviewing our responses.

Figure 1 and table 1 must be included in the Results section, as indicated by the CONSORT statement guidelines for RCTs (https://www.consort-statement.org/).

Response: We have moved Figure 1 and Table 1 to the results section (see page 6 & 7). 

Regarding comments 19 and 20, this reviewer fully agrees with the authors. Please, add in the text the references provided in the cover letter.

Response: The references have been included in section Brain Stimulation (see page 5, line 197).